# Urinary Tract Infections with Carbapenem-Resistant *Klebsiella pneumoniae* in a Urology Clinic—A Case-Control Study

**DOI:** 10.3390/antibiotics13070583

**Published:** 2024-06-24

**Authors:** Viorel Dragos Radu, Radu Cristian Costache, Pavel Onofrei, Adelina Miron, Carina-Alexandra Bandac, Daniel Arseni, Mihaela Mironescu, Radu-Stefan Miftode, Lucian Vasile Boiculese, Ionela-Larisa Miftode

**Affiliations:** 1Department of Urology, Faculty of Medicine, University of Medicine and Pharmacy “Gr. T. Popa”, 700115 Iasi, Romania; radu_ouatu@email.umfiasi.ro (V.D.R.); radu.costache@umfiasi.ro (R.C.C.); adelina.miron@umfiasi.ro (A.M.); 2Department of Urology and Renal Transplantation, “C.I. Parhon” University Hospital, 700115 Iasi, Romania; carina_bandac@email.umfiasi.ro (C.-A.B.); daniel_arseni@email.umfiasi.ro (D.A.); dr_mironescu@spitalulparhoniasi.ro (M.M.); 3Department of Morpho-Functional Sciences II, Faculty of Medicine, University of Medicine and Pharmacy “Gr. T. Popa”, 700115 Iasi, Romania; 4Department of Urology, Elytis Hope Hospital, 700010 Iasi, Romania; 5Department of Internal Medicine I, Faculty of Medicine, University of Medicine and Pharmacy “Gr. T. Popa”, 700115 Iasi, Romania; radu-stefan.miftode@umfiasi.ro; 6Department of Preventive and Interdisciplinarity, Medical Informatics and Biostatistics, Faculty of Medicine, University of Medicine and Pharmacy “Gr. T. Popa”, 700115 Iasi, Romania; vasile.boiculese@umfiasi.ro; 7Department of Infectious Diseases (Internal Medicine II), Faculty of Medicine, University of Medicine and Pharmacy “Gr. T. Popa”, 700115 Iasi, Romania; ionela-larisa.miftode@umfiasi.ro

**Keywords:** carbapenemases, carbapenem-resistant, *Klebsiella pneumoniae*

## Abstract

Background: The aim of our study was to analyze the factors associated with the increased risk of urinary tract infection (UTI) with carbapenem-resistant (CR) *Klebsiella pneumoniae* (*Kpn*) and the antibiotic resistance spectrum of the strains in patients. As secondary objectives, we elaborated the profile of these patients and the incidence of different types of carbapenemases. Methods: We conducted a retrospective case-control study in which we compared a group of 62 patients with urinary tract infections with CR *Kpn* with a control group consisting of 136 patients with urinary tract infections with multidrug-resistant (MDR), but carbapenem-sensitive (CS), *Kpn*, who were hospitalized between 1 January 2022 and 31 March 2024. Results: Compared to patients with urinary tract infections with CS *Kpn*, patients with urinary tract infections with CR *Kpn* were preponderant in rural areas (62.9% vs. 47.1%, *p* = 0.038) and more frequently had an upper urinary tract infection (69.4% vs. 36.8%, *p* < 0.01). Among the risk factors examined, patients in the study group had a higher presence of urinary catheters inserted for up to one month (50% vs. 34.6%, *p* = 0.03), rate of hospitalization in the last 180 days (96.8% vs. 69.9%, *p* < 0.01) and incidence of antibiotic therapy in the last 180 days (100% vs. 64.7%, *p* < 0.01). They also had a higher rate of carbapenem treatment in the last 180 days (8.1% vs. 0%, *p* < 0.01). Patients in the study group had a broader spectrum of resistance to all antibiotics tested (*p* < 0.01), with the exception of sulfamethoxazole–trimethoprim, where the resistance rate was similar in both groups (80.6% vs. 67.6%, *p* = 0.059). In the multivariate analysis, transfer from other hospitals (OR = 3.51, 95% and CI: 1.430–8.629) and treatment with carbapenems in the last 180 days (OR = 11.779 and 95% CI: 1.274–108.952) were factors associated with an increased risk of disease compared to the control group. The presence of carbapenemases was observed in all patients with CR *Kpn*, in the order of frequency New Delhi metallo-ß-lactamase (NDM) (52.2%), *Klebsiella pneumoniae* carbapenemase (KPC) (32.6%), and carbapenem-hydrolyzing oxacillinase (Oxa-48) (15.2%). Conclusions: The environment of origin and previous treatment with carbapenems appear to be the factors associated with an increased risk of urinary tract infection with CR *Kpn* compared to patients with urinary tract infections with CS *Kpn*. CR *Kpn* exhibits a broad spectrum of antibiotic resistance, among which is resistance to carbapenem antibiotics.

## 1. Introduction

In recent decades, more and more cases of CR Enterobacteriaceae infections have been reported worldwide [1,2]. The production of enzymes called carbapenemases is the main cause of resistance of Enterobacteriaceae to carbapenem antibiotics [3,4]. The majority of infections with Enterobacteriaceae that are CR are found in the urinary tract [5]. *Kpn* is one of the most common Gram-negative bacteria that are CR and cause urinary tract infections [2,6,7,8], although the reason for its prevalence compared to other Enterobacteriaceae is not known [9]. In South-Eastern Europe, *Kpn* is the most common CR Enterobacteriaceae, more common than *E. coli*, and thus plays an important role in the spread of antibiotic resistance [10]. 

These MDR infections are most commonly hospital-acquired infections associated with medical procedures [11,12]. *Kpn* is a commensal pathogen that occurs in the gastrointestinal tract, oral cavity, and nasopharynx [13], and causes 10% of nosocomial infections [14]. *Kpn*, including CR *Kpn*, can also be found in non-clinical environments, such as meat, seafood and vegetable samples [15]. In recent years, most extraintestinal infections with *Kpn* are multidrug-resistant [16]. Numerous risk factors for the occurrence of urinary tract infections with CR *Kpn* have been investigated, many of which are consistent with the risk factors for MDR urinary tract infections with Enterobacteriaceae. These include previous hospitalization, prolonged antibiotic treatment [4,17], comorbidities and invasive procedures [18,19,20], previous carbapenem use, number of patients admitted in the intensive care units (ICU) [20,21,22], presence of urinary catheters [19,23], previous colonization [23,24,25], and the formation of a bacterial biofilm on both the urothelial mucosa and abiotic surfaces such as urinary catheters [26,27,28,29].

Urinary tract infections with CR *Kpn* require prolonged antibiotic therapy and have a slow evolution and increased mortality [4,7]. However, to our knowledge, no study of this infection has been conducted in a urology clinic, where there are likely to be many patients with this disease. Most studies have been conducted in general hospitals or intensive care units [4,6,22,30,31]. It is not known what the patient profile and risk factors are for the occurrence of urinary tract infections with CR *Kpn* compared to urinary tract infections with MDR *Kpn* sensitive to carbapenems in a urology clinic. There are no current studies on the type of surgical procedures that favor the occurrence of urinary tract infections with CR *Kpn* in a urology clinic. Also, the sensitivity spectrum of germs is constantly changing and there are no recent studies in our geographic area that have analyzed this.

In recent years, we have observed an increase in the frequency of urinary tract infections with CR *Kpn* in our clinic. Therefore, we conducted a case-control study in which we analyzed the risk factors for the occurrence of urinary tract infections with CR *Kpn* compared to urinary tract infections with MDR *Kpn* sensitive to carbapenems and the resistance spectrum of the bacterium. We also analyzed the profile of patients in the two groups, the sensitivity spectrum of *Kpn* in the two groups, and the presence of enzymes conferring resistance to carbapenems in the case of the study group.

Our study is of interest not only to urologists, but also to infectious disease doctors, internal medicine doctors, and intensive care doctors, who treat this type of patient, and it is also a contribution to a better understanding of the worldwide spread of this type of germ, known for its wide distribution. It also helps in understanding of circulation of *Kpn* bacteria that possess enzymes that confer resistance to carbapenems.

## 2. Results

Between 1 January 2022 and 31 March 2024, there were 198 patients with urinary tract infections with MDR *Kpn* who met the inclusion criteria. Among them, 62 patients with UTIs with CR *Kpn* formed the study group, and the remaining 136 patients had UTIs with MDR *Kpn* that was carbapenem-sensitive. Table 1 shows the demographics of the patients in the two groups, the presence of comorbidities, urosepsis on admission, mortality rate, and transfer from other hospitals.

Patients in both groups had an average age of over 60 years, with the same frequency in men (59.7% vs. 53.7%, *p* = 0.43) and women (40.3% vs. 46.3% *p* = 0.52). Patients in the study group had a statistically significantly higher incidence of rural origin (62.9% vs. 47.1%, *p* = 0.038) and urinary tract infection at the renal level (69.4% vs. 36.8%, *p* < 0.01), compared to the control group. Regarding comorbidities, both groups had an increased incidence of DM (33.9% vs. 26.5%, *p* = 0.28), neoplasia (46.8% vs. 36%, *p* = 0.15), anemia (60.7% vs. 58.1%, *p* = 0.73), and obesity (24.2% vs. 20.6%, *p* = 0.56), but without significant statistical differences. Instead, we observed an increased incidence in favor of the study group in terms of the presence of renal failure (67.7% vs. 50.7%, *p* = 0.025) and neurological disorders (19.4% vs. 7.4%, *p* = 0.013). In the control group, the proportion of patients with arterial hypertension was higher (69.1% vs. 50%, *p* = 0.01).

Although the rate of presence of urosepsis on admission and mortality was higher in percentage terms in the study group, there were no statistically significant differences between these variables. We also note that there was a higher incidence of cases transferred from other hospitals in the study group (24.2% vs. 8.8%, *p* < 0.01).

Table 2 shows the presence of urinary catheters at the time of diagnosis, according to the type and duration of their presence until the time of diagnosis of urinary tract infection with *Kpn* in the two groups.

We observed a very high percentage of patients with catheters at the time of diagnosis in the two groups (88.71% vs. 83.1%, *p* = 0.26), most frequently double-J catheters (46.8% vs. 36.8%, *p* = 0.18) and Foley urethral catheters (29% vs. 25.7%, *p* = 0.62), but also nephrostomy catheters, cystostomy catheters, and cutaneous ureterostomy catheters, with no statistically significant differences between the two groups. Instead, patients in the study group had a higher incidence of catheters inserted earlier than one month compared to the control group (50% vs. 34.6%, *p* = 0.03).

Table 3 shows the comparisons of the frequency of occurrence of the investigated risk factors.

Patients with urinary tract infections with CR *Kpn* had a higher rate of hospitalization and antibiotic therapy in the last 180 days compared to the control group (*p* < 0.01). They also had more days of hospitalization (*p* = 0.034). Regarding the type of urological procedures, patients in the study group had more insertions/replacements of nephrostomy catheters (*p* < 0.01), but fewer insertions/replacements of Foley urethral catheters (*p* < 0.01) and percutaneous nephrolithotomies (*p* = 0.037).

Moreover, all types of interventions, including the insertion and replacement of ureteral double-J catheters occurred in similar, statistically insignificant percentages. However, it should be noted that almost half of the patients in both groups had a history of double-J catheter insertion or replacement (46.8% and 38.2%, respectively, *p* = 0.25).

Multivariate analysis showed that for patients, transfer from other hospitals and previous carbapenem treatment were factors associated with an increased risk of CR *Kpn* UTI compared to CS *Kpn* UTI: OR = 3.512 (95% CI: 1.430–8.629) and OR = 11.779 (95% CI: 1.274–108.952). On the other hand, patients wearing a Foley urethral catheter had a lower risk of CR *Kpn* UTI (OR = 0.164).

Table 4 shows the frequency of antibiotic resistance in the two groups. The first 14 antibiotics were tested in all patients enrolled in the study.

We note that in addition to resistance to carbapenems, patients in the study group had a urinary tract infection with *Kpn* with an extended spectrum of resistance (statistically significant) to all commonly used antibiotics compared to the control group, with the exception of sulfamethoxazole–trimethoprim, where resistance occurred in approximately the same percentage (80.6% vs. 67.6% *p* = 0.059).

All patients in the study group showed resistance to meropenem, and 55 of 62 patients (90.3%) showed resistance to imipenem. Colistin was tested in 43 patients, of whom 13 (30%) showed resistance. Tigecycline was tested in four patients, all with resistance. Aztreonam was tested in six patients, and four showed resistance. Ceftazidime/avibactam was tested in five patients, and four showed resistance. Cefpozolam/tazobactam was tested in five patients, all with resistance. Imipenem/cilastatin/relebactam was tested in five patients, three of whom showed resistance. Four patients showed resistance to all 14 commonly tested antibiotics, including colistin. One case showed resistance to all 14 routinely tested antibiotics, including colistin, tigecycline, and aztreonam.

Table 5 shows the type of enzymes that confer *Kpn* resistance to carbapenems for the isolates of 30 patients examined. 

Of the five enzymes tested, three were highlighted, and all isolates of the 30 patients presented one or two enzymes. The most common was NDM (38.7%), followed by KPC (24.2%) and Oxa-48 (11.3%). In 25.8% of patients, i.e., more than a quarter, *Kpn* with two enzymes was detected.

## 3. Discussions

Patients with CR *Kpn* had a higher incidence of upper urinary tract infections and a higher rate of patients transferred from other hospitals than patients with CS *Kpn*. Patients from both groups had a high incidence of the presence of urinary catheters of any type, but in patients with CR *Kpn*, catheters inserted earlier than one month predominated. Patients with CR *Kpn* had a history of urologic procedures; many of them remained with permanent urinary catheters, had a higher rate of carbapenem treatment, and more hospital days compared to patients from the control group. Patients in both groups showed increased resistance to the antibiotics commonly used in urology. The resistance to carbapenems is due to the presence of NDM, KPC, and Oxa-48 enzymes.

To our knowledge, there are few internation and local studies in the literature that analyze the profile of patients with CR *Kpn* UTIs in a urology clinic, as well as the risk factors and spectrum of bacterial sensitivity in this particular group of patients [32,33,34]. The main limitation of the study is its retrospective nature. Another limitation of our study is that we analyzed the presence of enzymes conferring resistance to carbapenems in only 30 of the 62 patients in the study group. In addition, some patients who underwent surgery in our clinic were monitored in other departments, and it is likely that those diagnosed with CR *Kpn* UTIs were treated in other urology or infectious disease departments. Conversely, a significant percentage of patients with CR *Kpn* UTIs were transferred from other departments, making it difficult to accurately determine the incidence of this infection in our clinic. However, this does not affect the identification of patient profiles, factors associated with increased risk, and the spectrum of sensitivity. Another potential bias is that we could not determine the number of previous hospitalizations for each patient, although we know that each hospitalization can increase the risk of MDR-UTI. 

As expected, the profile of the CR *Kpn* UTI patients is similar to that of the control group. The predominant age is over 60 years with a higher mortality potential [35,36], as most patients requiring urological interventions are elderly. Although women are in principle more susceptible to urinary tract infections, we did not find any difference between genders in our study. The localization of urinary tract infections in the upper urinary tract is explained by the higher incidence of double-J ureteral catheters and nephrostomy catheters in the study group patients, while in the control group there was a greater number of patients with a permanent urethral catheter. The presence of a large number of comorbidities indicates the sensitivity of these patients to develop UTIs by lowering the body’s immunity. DM, neoplasia, and anemia are known factors for reducing host defenses against infection, but they did not appear to be factors that increased the incidence of CR *Kpn* UTI. The fact that there were a greater number of patients with renal failure in the study group is explained by the likely high incidence of patients with sepsis and secondary organ failure. We mention that we could not determine whether the patients had normal renal function before the UTI, so we could not assess whether the renal failure was pre-existing or secondary to sepsis, so we cannot conclude that renal failure could increase the risk of UTI with CR *Kpn*. Neurological disorders were the only comorbidities that occurred more frequently in the study group compared to the control group. The fact that patients in both groups had a high rate of urosepsis on admission, as in other studies [37], demonstrates the aggressiveness of MDR *Kpn*. However, mortality was lower, demonstrating the better prognosis of urosepsis of urinary origin.

An interesting result of our study was the higher incidence of patients transferred from other hospitals in the study group in multivariate analysis. These findings can be explained by the aggressiveness of the infection and perhaps the limited possibilities of antibiotic treatment in other hospitals, as well as the need for urgent urological intervention, which is not possible in other hospitals. However, the occurrence of CR *Kpn* in other secondary hospitals in the north-eastern region of Romania indicates a wide distribution and the fact that this resistance to carbapenems may occur in several regions probably after similar treatments with antibiotics.

As in other studies [18,19,20,38,39,40], as expected, antibiotic therapy and previous hospitalization together with urological procedures were predominant in both groups of patients, showing a high probability that these infections were nosocomial hospital infections, especially since colonization in the general population with CR *Kpn* has not been reported in our country. However, the statistically significant higher rate in the study group shows that these factors intervene to a greater extent in the occurrence of urinary tract infections with CR *Kpn*. The ICU admissions were similar in both groups, occurring in patients previously diagnosed with CR *Kpn* and CS *Kpn* UTIs, so we could not prove that ICU admissions were risk factors in the case of our group, as in other studies [17,22]. Although we did not have the opportunity to find out how many previously operated patients were hospitalized in the ICU, most of the patients described in our study who underwent this type of surgery were not routinely hospitalized in the ICU.

Previous treatment with carbapenems has been shown to be associated with an increased risk of urinary tract infection with CR *Kpn*, even in multivariate analysis. However, in percentage terms, this factor occurred in less than 10% of patients, which indicates that other factors are also involved in the occurrence of resistance to carbapenems.

The insertion/replacement of double-J catheters were the most common operations performed in both groups before the occurrence of MDR-UTI although these procedures were frequently performed as emergencies in our clinic. Percutaneous nephrostomy was performed more frequently in the study group, an operation that was also predominantly performed as an emergency, as was the insertion of double-J catheters for obstructive pyelonephritis, i.e., in patients who already had a urinary tract infection and who were administered antibiotics, most frequently injectable cephalosporins, sometimes over long periods of prolonged hospitalization, which probably contributed to the emergence of bacterial resistance. On the other hand, in patients with CS *Kpn* UTIs, Foley urethral catheters were inserted and changed more frequently, resulting in a higher lower tract UTI incidence, thus explaining the lower number of hospital days and shorter duration of antibiotic treatment. At the same time, only a few of these patients were admitted to hospital, as this procedure was performed on an outpatient basis. This explains the lower incidence of urinary tract infections in these patients with CR *Kpn* due to the lower rate of hospitalization, the less frequent antibiotic therapy, and the shorter duration of antibiotic treatment. Although we observed a higher rate of percutaneous nephrolithotomy in the control group, we cannot say that this type of surgery has a lower risk for the subsequent occurrence of urinary tract infections with CR *Kpn*, probably because the shorter hospital stay and antibiotic therapy as well as the rapid extraction of the nephrostomy tube after surgery explain this.

As in other local studies [34,41,42] showing the presence of urinary catheters in association with *Kpn* UTIs, the current study also found a high presence of patients with urinary catheters at the time of diagnosis. Numerous studies have demonstrated bacterial colonization of catheters, including with *Kpn* through its ability to bind to biotic and abiotic surfaces via fimbriae [43]. After fixation, the germs form a bacterial biofilm [26,29,37,44], an element of bacterial persistence and the development of resistance to carbapenems [27]. We have shown that not only does the presence of Foley urethral catheters confer a risk of urinary tract infection with CR *Kpn* [6], but also the presence of double-J ureteral catheters, nephrostomy, cystostomy, or ureterostomy double-J catheters, as in other studies [26,27,29].

The actual incidence of patients with permanent urethral catheters and urinary tract infections with CR *Kpn* is therefore likely to be higher. The fact that patients with UTIs with CR *Kpn* were more likely to have catheters installed for less than one month indicates that infection is likely to occur more rapidly and perhaps more frequently than in patients with urinary catheters but colonized with CS *Kpn*, even if they are also MDR, due to increased aggressiveness after colonization. Our results suggest that all measures should be taken to reduce the indication of chronic wear of urinary catheters, especially double-J ureteral catheters, and to shorten the time to their removal if they do need to be inserted. CR *Kpn* UTI was obviously a nosocomial infection, even if the diagnosis was made at the time of readmission. In patients discharged without indwelling urinary catheters, the infection rate was low, although patients were exposed to the same risk of infection in hospital as patients with urinary catheters. In other local studies, we have not reported the occurrence of MDR UTIs in patients discharged without urinary catheters [45]. It is likely that a normal urinary system without a urinary catheter does not develop UTIs despite possible contamination with CR *Kpn*, which is why we do not suggest screening all discharged patients for MDR UTIs as in other studies [46], but only those patients discharged with urinary catheters, especially if their removal is not expected for a long time.

Regarding antibiotic resistance, we note that in addition to the presence of resistance to carbapenems, the study group also had a higher rate of resistance to quinolones and cephalosporins, and frequently to almost all antibiotics tested from these two groups, suggesting that the occurrence of carbapenem resistance depends on the same favorable factors as the occurrence of resistance to other antibiotics. The only exception was sulfamethoxazole–trimethoprim, to which resistance was similar in both study groups. We note that, in contrast to other studies that reported increased resistance to imipenem as well as the presence of the enzyme imipenem-hydrolyzing metallo-beta-lactamases (IMPs) [47], in our study group all patients had resistance to meronem and not all patients to imipenem, without objectifying the presence of IMPs.

This can be explained by the predominant use of meronem for the treatment of carbapenem-sensitive MDR UTIs in our clinic, whereby the increased use led to the occurrence of resistance [48]. Antibiotic resistance was very different in the two groups, indicating the simultaneous presence of numerous serotypes of *Kpn* in the clinic. A favorable element for treatment options was the fact that a relatively high percentage of patients in the study group had a urinary tract infection with *Kpn* with preserved sensitivity to gentamicin, amikacin, fosfomycin i.v., or trimethoprim/sulfamethoxazole. Although it is considered a second-line drug [49], colistin had a high resistance rate, which is explained by the fact that it was frequently used in our clinic to treat urinary tract infections with CR Enterobacteriaceae and that resistance to this antibiotic occurs relatively quickly [46]. Of concern is the presence of resistance to last-generation antibiotics that were not as commonly used in the clinic, such as ceftozolan/tazobactam, ceftazidime/avibactam, or imipenem/cilastatin/relebactam, demonstrating the broad resistance of CR *Kpn* to multiple antibiotics. Despite the aggressiveness of the germs, however, mortality was low, which indicates a mostly favorable evolution of the infection with CR *Kpn* at the urinary tract level.

Regarding the mechanism by which *Kpn* acquires resistance to carbapenems, we were able to demonstrate that the most important mechanism is the secretion of carbapenemases. The enzymes we identified are NDM, KPC, and Oxa-48, with some germs even presenting two enzymes. Although the presence of KPC was first reported in the United States and the presence of NDM in Asia [47], their identification in our country demonstrates the virtually worldwide distribution of *Kpn* genes resistant to carbapenems. 

CR *Kpn* is constantly on the rise in hospitals and in the north-eastern region of Romania, and the occurrence of pan-drug resistant germs will pose serious problems for the public health system in our country. Therefore, the prudent use of antibiotics by shortening the duration of administration and the fastest possible removal of urinary catheters, as well as the shortening of waiting times of patients with indwelling urinary catheters until definitive endoscopic urological treatment, are proposals to limit the transmission of these germs, including carbapenem-sensitive MDR germs, and further directions of study.

## 4. Materials and Methods

### 4.1. Patient Selection and Data Collection

We conducted a retrospective case-control study in patients diagnosed with MDR UTIs with *Kpn* between 1 January 2022 and 31 March 2024 at the urology clinic of the Teaching Hospital “Dr. C.I. Parhon” in Iasi, Romania. The patient data were retrieved from the electronic documents of the hospital using the ICD-10 code from the International Classification of Diseases, 10th edition. We identified all patients who had the code B96.1 (*Klebsiella pneumoniae*, the cause of some diseases classified in other chapters). It should be mentioned that all identified patients had a urinary tract infection. There was a single case associated with a urinary tract infection and colonization with *Kpn* in the tracheal aspirate, with the bacteria having the same sensitivity spectrum. As they did not have an antibiogram in the electronic data, 4 patients were excluded from the study. The 214 identified patients were divided into two groups based on the antibiogram prepared in the hospital’s microbiology laboratory, depending on whether resistance to carbapenems was present. The study group consisted of all patients with *Kpn* urinary tract infections resistant to at least one carbapenem, defined as CR [50], and the rest of the patients had *Kpn* urinary tract infections sensitive to all tested carbapenems.

Among the latter, there were 14 patients with a urinary tract infection with multidrug-sensitive *Kpn* who were excluded from the study, as well as two cases with a urinary tract infection with MDR *Kpn* and an infection associated with COVID-19, as we were unable to determine whether the infection with *Kpn* occurred before that with COVID-19 so we could consider the COVID-19 infection as a risk factor. The remaining patients with urinary tract infections with MDR *Kpn* who showed preserved sensitivity to carbapenems formed the control group. We defined MDR as all germs that showed resistance to at least one antibiotic from at least 3 different classes [51]. For the diagnosis of urosepsis, we used the definition from the third international consensus definitions for sepsis and septic shock (Sepsis 3) [52].

### 4.2. Microbiology Methods

A urinary tract infection was diagnosed in the presence of a positive urine culture and clinical and paraclinical phenomena suggestive of a urinary tract infection (pollakiuria, urinary urge, pyuria or cloudy urine due to leukocytes, dysuria, febrile syndrome, lower back pain, and leukocytosis). We defined a positive urine culture if it was above 10^5^ UFC/mL. Antimicrobial susceptibility testing was performed by the disk diffusion method using the EUCAST clinical breakpoint for interpretation of minimum inhibitory concentrations and zone diameters.

The following antibiotic discs (producer: MAST GROUP LTD, Mast House, UK) were used: ampicillin (10 µg), amoxicillin–clavulanic acid (10–20 µg), cefuroxime (30 µg), ceftriaxone (30 µg), ceftazidime (10 µg), cefepime (30 µg), piperacillin–tazobactam (6–30 µg), ciprofloxacin (5 µg), levofloxacin (5 µg), imipenem (10 µg), meropenem (10 µg), gentamicin (10 µg), amikacin (30 µg), fosfomycin (200 µg), aztreonam (30 µg), nitrofurantoin (100 µg), trimethoprim–sulfamethoxazole (1.25–23.75 µg), ceftazidime–avibactam (4–10 µg)—ZAFICEFTA^®^, ceftozolan-tazobactam (10–30 µg)—ZERBAXA^®^, and imipenem/cilastatin/relebactam (10–25 µg)—RECARBIO^®^.

The results for sensitivity to tigecycline and colistin were reported after testing with the Microscan WalkAway DxM 1040 device (Beckman Coulter, Indianapolis, IN, USA) and we obtained the minimum inhibitory concentration (MIC) according to the EUCAST criteria; for tigecycline MIC < 0.5 mg/L sensitive and for colistin, MIC < 2 mg/L sensitive.

If the result of the antibiogram showed partial (intermediate) resistance, we considered resistance to this antibiotic. It should be mentioned that we did not have a single patient with a positive urine culture who did not have at least one clinical or paraclinical sign of a UTI. We did not have a single patient with *Kpn* in whom the urine culture was <10^5^ UFC/mL. Urine cultures were obtained in the first hours after admission, before initiation of empirical antibiotic therapy and during hospitalization for suspected UTI. In 3 patients from the study group who had negative urine cultures on admission, the diagnosis of CR *Kpn* UTI was established by repeat urine cultures on days 9, 12, and 14.

All other patients in the study group were diagnosed by urine culture performed at the time of admission. In the control group, urinary tract infection with CS *Kpn* was diagnosed in 4 patients by urine cultures performed on days 5, 7, 12, and 15 after admission. The remaining patients in the control group were diagnosed by urine culture performed at the time of admission. The antibiogram was used to test for resistance to a range of 14 antibiotics in all patients in the study. In the study group, if resistance to carbapenems was detected or suspected, resistance to other antibiotics was also tested for at least one of the following antibiotics for each patient: colistin, aztreonam, tigecycline, ceftazidime/avibactam, cefpozolam/tazobactam, and imipenem/cilastatin/relebactam.

Isolates resistant to at least one carbapenem (according to the EUCAST criteria) were tested for the production of carbapenemases using the immunochromatographic test NG-Test CARBA5 (NG Biotech, Guipry, France), a rapid test that can detect 5 types of carbapenemases simultaneously: KPC (*Kpn* carbapenemase), NDM (New Delhi metallo-beta-lactamase), OXA-48 (oxacillinase), VIM (Verona integron-encoded metallo-beta-lactamase), and IMP (imipenemase, or IMP metallo-beta-lactamase). The last 30 patients with CR *Kpn* urinary tract infections enrolled in the study were tested for the presence of carbapenemases. We mention that we had no pregnant women and patients under the age of 18 in either group.

### 4.3. Outcome Measures

We compared the two patient groups in terms of gender, age, residence, type of urinary tract infection—upper or lower urinary tract—and comorbidities: presence of diabetes, neoplasia, renal and heart failure, anemia, neurological disease, hypertension, and obesity. We also analyzed the presence of urosepsis on admission, the number of patients transferred from other hospitals, and mortality in the two groups. All comorbidities considered in the study favor the occurrence of UTIs by lowering the body’s defense mechanisms or favoring the occurrence of some urological diseases that predispose to UTIs, such as the occurrence of renal lithiasis in immobilized patients with neurological diseases. As our hospital is a teaching hospital, it receives complex cases from other hospitals. For this reason, we analyzed the presence of urinary tract infections with *Kpn* before transferring from other hospitals in order to have a perspective on this infection at the level of the entire north-eastern region of Romania. The presence of urosepsis and mortality gives us information about the severity of urinary tract infection with *Kpn* in comparison of the two groups.

With regard to potential risk factors, we analyzed the presence of urinary catheters at the time of diagnosis, a recognized factor for increasing the incidence of MDR UTI [53], such as urethral catheters, double-J ureteral catheters, percutaneous nephrostomy catheters, cystostomy catheters, and percutaneous ureterostomy catheters. Regarding the presence of urinary catheters, we analyzed patients with catheters inserted earlier than one month ago and over one month separately in the two groups, knowing that the presence of catheters is a risk factor for bacterial colonization and the subsequent occurrence of infections. Other risk factors that were analyzed were hospitalization in the last 180 days, antibiotic therapy in the last 180 days, the percentage of patients who required hospitalization in the ICU, carbapenem therapy in the last 180 days, and the types of surgical procedures that preceded the UTI with *Kpn* and were performed up to 180 days before the UTI was diagnosed. All of these factors are recognized to promote the emergence of bacterial resistance, including the occurrence of UTIs with CR Enterobacteriaceae. We analyzed the resistance spectrum of *Kpn* in the two groups and the presence of the enzymes responsible for the resistance of *Kpn* to carbapenems in the last 30 patients of the study group.

### 4.4. Statistical Analysis

To describe the data statistically, we computed absolute and relative frequencies for categorical variables, while the mean and standard deviation were used for numerical variables. To compare the groups (univariate analysis) on the basis of statistical hypothesis, the *t*-test was used for the mean and the chi-square test for the frequencies. Fisher’s exact test (superscript F) was only used if chi-square consistency was not met. For primary data collection, we used Microsoft Excel version 2016 (Microsoft Corporation, Redmond, WA, USA), while data analysis was performed using SPSS version 24 (IBM, Armonk, NY USA).

## 5. Conclusions

The patients with CR *Kpn* UTIs were over 60 years of age and had multiple comorbidities, particularly neoplasia, DM and anemia. All patients with CR *Kpn* UTIs admitted to the urology clinic underwent urologic surgery, predominantly endoscopic, and had a history of antibiotics, with the most common surgeries being those in which patients remained urinary catheter carriers. Transfer from other hospitals and previous treatment with carbapenems appear to be factors associated with an increased risk of CR *Kpn* UTIs compared to patients with CS *Kpn* UTIs. Compared to the control group, the patients in the study group have an extended resistance spectrum that is not limited to resistance to carbapenems. The main pathway of *Kpn* resistance to carbapenems in the urologic clinic is the presence of carbapenemases and probably the formation of a bacterial biofilm on the surface of urinary catheters. Future prospective studies are needed to validate our results.

## Figures and Tables

**Table 1 antibiotics-13-00583-t001:** Characteristics of the patients in the two groups.

		Study Group (*n =* 62)	Control Group (*n* = 136)	*p*-Value for Chi-Square Test
Male		37 (59.7%)	73 (53.7%)	0.43
Female		25 (40.3%)	63 (46.3%)	
Age (mean ± SD)	Male	65.84 ± 14.09	67.56 ± 12.98	0.52 ^t^
	Female	64 ± 16.97	60.95 ± 16.25	0.46 ^t^
Residence area (U)		23 (37.1%)	72 (52.9%)	0.038
Localization of UTI	Upper UTI	43 (69.4%)	50 (36.8%)	<0.01
	Lower UTI	19 (30.6%)	86 (63.2%)	
Comorbidities				
Type 2 diabetes (DM)		21 (33.9%)	36 (26.5%)	0.28
Neoplasia		29 (46.8%)	49 (36%)	0.15
Kidney failure		42 (67.7%)	69 (50.7%)	0.025
Heart failure		12 (19.4%)	36 (26.5%)	0.27
Anemia		37 (60.7%)	79 (58.1%)	0.73
Stroke sequelae (neurological disorders)		12 (19.4%)	10 (7.4%)	0.013
Hypertension		31 (50%)	94 (69.1%)	0.01
Obesity		15 (24.2%)	28 (20.6%)	0.56
Urosepsis at the moment of admission		32 (51.6%)	57 (41.9%)	0.20
Mortality		5 (8.1%)	4 (2.9%)	0.14 ^F^
Transfer from other hospitals		15 (24.2%)	12 (8.8%)	<0.01

t—Student’s *t*-test; SD—standard deviation; F—Fisher’s exact test.

**Table 2 antibiotics-13-00583-t002:** Urinary catheters at the time of diagnosis.

		Study Group (*n* = 62)	Control Group (*n* = 136)	*p*-Value for Chi-Square Test
Permanent urethral catheter		18 (29%)	35 (25.7%)	0.62
Permanent double-J ureteral catheter		29 (46.8%)	50 (36.8%)	0.18
Permanent nephrostomy catheter		8 (12.9%)	25 (18.4%)	0.33
Permanent ureterostomy double-J catheter		1 (1.6%)	1 (0.7%)	0.56 ^F^
Permanent cystostomy catheter		2 (3.2%)	5 (3.7%)	0.99 ^F^
Total permanent urinary catheters		58 (3 patients had 2 urinary catheters) 93.5%	116 (3 patients had 2 urinary catheters) 85.2%	0.09
Total number of patients with permanent urinary catheters	<1 month	31 (50%)	47 (34.6%)	0.03
	>1 month	24 (38.71%)	66 (48.5%)	0.23
Total number of patients with permanent urinary catheters		55 (88.71%)	113 (83.1%)	0.26
Total number of patients without urinary catheter at the time of hospitalization		7 (11.29%)	23 (16.9%)	0.26

F—Fisher’s exact test.

**Table 3 antibiotics-13-00583-t003:** Risk factors of CR *Kpn* UTI.

	Study Group (*n* = 62)	Control Group (*n* = 136)	*p*-Value for Chi-Square Test
Hospitalization in the past 180 days	60 (96.8%)	95 (69.9%)	<0.01
Antibiotherapy in the past 180 days	62 (100%)	88 (64.7%)	<0.01
Hospitalization days (mean ± standard deviation)	10.34 ± 9.73	7.48 ± 5.59	0.034
ICU stay (number of patients)	7 (11.3%)	8 (5.9%)	0.18
Previous history of carbapenems treatment	5 (8.1%)	0 (0%)	<0.01 ^F^
Types of urological interventions performed before the diagnosis of MDR infections			
TURP (+/− lithotripsy)	2 (3.2%)	5 (%)	1 ^F^
TURBT	7 (11.3%)	8(5.8%)	0.18
Percutaneous nephrostomy tube insertion/replacement	8 (12.9%)	2 (1.4%)	<0.01
Urethral catheter insertion/replacement	4 (6.5%)	38 (27.9%)	<0.01
Percutaneous nephrolithotomy (PCNL)	5 (8.1%)	27 (19.9%)	0.037
Double-J catheter insertion/replacement	29 (46.8%)	52 (38.2%)	0.25
Open surgery	4 (6.5%)	4 (2.9%)	0.26 ^F^
Ureterostomy double-J catheter replacement	1	0	
Cystostomy	2 (3.23%)	0 (0%)	0.09
Total patients with urological maneouvers, before the occurrence of MDR	62	136	

F—Fisher’s exact test.

**Table 4 antibiotics-13-00583-t004:** Antibiotic resistance in the 2 groups.

	Study Group (*n* = 62)	Control Group (*n* = 142)	*p*-Value for Chi-Square Test
Ampicillin	62 (100%)	126 (92.6%)	<0.01
Amoxicillin/acid clavulanic	62 (100%)	102 (75%)	<0.01
Trimethoprim/sulfamethoxazole	50 (80.6%)	92 (67.6%)	0.059
Nitrofurantoin	60 (96.8%)	81 (59.6%)	<0.01
Ciprofloxacin	62 (100%)	106 (77.9%)	<0.01
Levofloxacin	62 (100%)	91 (66.9%)	<0.01
Cefuroxime	62 (100%)	89 (65.4%)	<0.01
Ceftriaxone	62 (100%)	81 (59.6%)	<0.01
Ceftazidime	62 (100%)	80 (58.8%)	<0.01
Cefepime	62 (100%)	75 (55.1%)	<0.01
Piperacillin/Tazobactam	59 (95.2%)	24 (17.6%)	<0.01
Imipenem	56 (90.3%)	0 (0%)	<0.01
Meropenem	62 (100%)	0 (0%)	<0.01
Gentamicin	52 (83.9%)	58 (42.6%)	<0.01
Colistin (tested for 43 patients)	13 (30%)	Not tested	
Fosfomycin	53 (85.5%)	Not tested	
Amikacin	51 (82.3%)	Not tested	
Nr. of cases resistant to one carbapenem	6 (9.67%)	0 (0%)	
Nr. of cases resistant to all carbapenem	56 (90.33%)	0 (0%)	

**Table 5 antibiotics-13-00583-t005:** Types of carbapenemases observed in the isolated specimens.

	Isolates Tested (*n* = 30)	Incidence (%)
*Kpn* enzymes		
Isolates with KPC	15	24.2%
Isolates with Oxa-48	7	11.3%
Isolates with NDM	24	38.7%
Isolates with 2 enzymes	16	25.8%
Total number of KPC, Oxa-48 and NDM detected	46	

## Data Availability

Data are contained within the article.

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
