# Peer review of "Urinary Tract Infections with Carbapenem-Resistant Klebsiella pneumoniae in a Urology Clinic—A Case-Control Study"

_antibiotics, 2024, doi:10.3390/antibiotics13070583_

Round 1
Reviewer 1 Report
Comments and Suggestions for Authors
The work entitled “Urinary tract infections with carbapenem-resistant Klebsiella pneumoniae in a urology clinic. A case-control study” provides a retrospective case-control analysis of putative factors influencing the development of carbapenem-resistant Klebsiella pneumoniae infections in patients hospitalized by previous urinary infections/other urinary diseases. The study brings novel data on clinical cases, some interesting relationships between putative causes of the multi-resistant bacterial infections and patient outcomes after carbapenems antibiotic treatment.
Nevertheless, the quality of the work doesn’t justify the number of contributing authors. Twelve authors for a paper based on a statistical analysis of hospital archives, seems excessive. Still in the mentioned author contributions, it is not clear to me how in this study you can clearly differentiate methodology from Investigation, or validation from data curation. Please move the authors that did not contribute decisively in the analysis of the data to the “acknowledgments” section.
Main suggestions:
Regarding what you mentioned in lines 196 to 198, if is is feasible could you please try to correlate bacterial resistance profile with the clinic of procedence of patients?
On your sentence in line 201 “...we know that each hospitalization can increase the risk of MDR-UTI.”: are data from bacteria in the clinical environment available? Does any center sample the walls, the air ducts, the ceiling, the lights, the beds for multi resistant bacteria? Any metagenomic study?
In line 213 you wrote: “The presence of a large number of comorbidities indicates the sensitivity of these patients to develop UTI by lowering the body's immunity.” Did you analyze a possible correlation between any immune parameter and MDR bacterial infection in patients?
Although it could be interesting for a board discussion, the main discussion in the manuscript is too long. For the sake of the readability, please shorten the 3 page discussion (i.e.: from line 236 to 358), to only the most relevant data, and sum up avoiding guesses and hypothesis when possible.
Please include some representative antibiograms in supplementary materials, if available.
Minor corrections:
In line 29 I suggest to use the words produced, elaborated, or determined instead of realized.
In lines 34 -35 please re-phrase as: urinary tract infections with CR Kl pn, were more preponderant in rural areas than infections with CS KL (62,9%, vs 47,1%, p=0.038).
In line45 acronyms should be clarified beforehand: what is NDM? What is KPC? Etc.
In line 69 please rephrase “Intensive care units (ICU) stay” as “time spent at ICU”
In Table 4 the carbapenem resistant numbers and percentages are misplaced in the wrong column.
In line 169 please write type instead of "case"
In Table 5 you wrote “Total number of enzymes detected”, please clarify what you mean with this. Did you detected 46 different carboxipeptidases?
In the “Materials and Methods” section please include brands of antibiotics used.
What you wrote from line 434 to line 446 doesn't belong to Materials and Methods.
The “Acknowledgments” section: please update or erase if there is no acknowledgments. Maybe you should acknowledge the nurses, the technicians and biochemistry personnel at the diagnosis bacterial culture lab, if not in the author list.
If all the corrections and suggestions are implemented, the paper should be published without further round of revisions.
My best,
The reviewer.
Reviewer 2 Report
Comments and Suggestions for Authors
The authors can give explanation for more number of control group compared to study group. Why?
What are all other possible infection along with UTI?
In conclusion they can refine better for clarity.
Reviewer 3 Report
Comments and Suggestions for Authors
In this study the authors analyse those factors associated with the increased risk of urinary tract infection by carbapenem-resistant Klebsiella pneumoniae strains. The antibiotic resistance spectrum of these isolates is also presented. To their knowledge this is the first study performed specifically at a urology clinic. However, a similar study with overlapping results is accessible in an available literature source: Carbapenem-resistant Klebsiella pneumoniae infection outbreak in a tertiary urology clinic: analysis of influencing factors with a controlled trial https://www.ncbi.nlm.nih.gov/pmc/articles/PMC7080388/.
Furthermore, two publications on carbapenem resistant K. pneumoniae are available from the authors country, one of them even form a tertiary urology clinic:
1. Carbapenemase-producing uropathogens in real life: epidemiology and treatment at a County Emergency Hospital from Eastern Romania. DOI 10.25122/jml-2023-0139
2. Characterization of the Mechanisms Underpinning Carbapenem Resistance in a Tertiary Urology Clinical Hospital - a Pilot Study. Romanian Journal of Urology nr. 4 / 2019 • vol 18. 22-25.
In this way –at least partly - the originality of the findings are questionable.
Remarks and questions:
1. The abbreviation KL. pn. has not been used in the literature. Instead Kpn is the usual form.
2. „…antibiotic resistance spectrum of these patients…”: the resistance refers not to the patients but to the strains isolated form their samples.
3. The authors state that rural setting, antibiotic therapy with carbapenems, and previous hospitalization promote the infection with carbapenem resistant K. pneumoniae. In rural settings without hospitalisation antibiotic pressure, and more specifically carbapenem selection pressure is not likely. Urine samples were taken within hours after admission, therefore the infections must have been acquired before the tertiary care. As patients with rural setting might be more probably hospitalised first in secondary care hospitals than immediately in tertiary care setting, the resistant pathogen might have been acquired during secondary care, and not at the place of rural residence. In this way the rural origin of their infections is questionable despite of their residence.
4. The Materials and Methods section is not structured into items like Patients and conditions, Bacterial isolates, etc., only the Statistical analysis is separated as a special point. The section is formed into a continual text starting with the patient aspects, continues with aspects of the pathogen, then describes the assays, and finally returns to patient conditions. The manufacturer of the antibiotic disks is not given.
5. It is stated in the text: The incidence of catheters inserted earlier than one month is significantly higher in the study group then in the control group. However, Table 2. shows this difference just for the less than one month group.
6. In Table 4. antibiotic resistances are shown. For imipenem and meropenem, as expected, high or total resistance are indicated for the study group, and no resistance is shown for the control group. On the contrary, the two last lines of the Table are empty for the study group, and resistance appears for the control group. How can it be interpreted?
7. Table 5. demonstrates the types of carbapenemases produced. Again here, not the patients but their isolates were tested. Out of the 62 patient isolates only 30 were included in this assay. The Table and the textual explanation are difficult to interpret together. Though 30 isolates were tested, the incidences in the Table are calculated for the 46 enzymes, or in one case for the total 62 isolate number in the test group. On the contrary, in the text the percentages are calculated for all the 62 isolates for all enzymes. What is the reason for that?
Comments on the Quality of English LanguageOnly minor corrections are needed.
Round 2
Reviewer 3 Report
Comments and Suggestions for Authors
x
The authors have observed all my comments. The suggested new references have been included. The quality of the manuscript has significantly improved.
Some minor comments:
Klebsiella pneumoniae and E. coli have to be italicized everywhere in the manuscript where applied.
Line 153 „others hospitals” the word „others) should stand in singular.
Author Response
The authors have observed all my comments. The suggested new references have been included. The quality of the manuscript has significantly improved.
Thank you
Some minor comments:
Klebsiella pneumoniae and E. coli have to be italicized everywhere in the manuscript where applied.
Corrected
Line 153 „others hospitals” the word „others) should stand in singular.
Corrected, thank you